# COVID-19 Related Early Google Search Behavior and Health Communication in the United States: Panel Data Analysis on Health Measures

**DOI:** 10.3390/ijerph20043007

**Published:** 2023-02-09

**Authors:** Binhui Wang, Beiting Liang, Qiuyi Chen, Shu Wang, Siyi Wang, Zhongguo Huang, Yi Long, Qili Wu, Shulin Xu, Pranay Jinna, Fan Yang, Wai-Kit Ming, Qian Liu

**Affiliations:** 1School of Management, Jinan University, Guangzhou 510632, China; 2College of Economics, Jinan University, Guangzhou 510632, China; 3School of Journalism, Fudan University, Shanghai 200433, China; 4Institute of Agricultural Resources and Regional Planning, Chinese Academy of Agricultural Sciences, Beijing 100081, China; 5Laboratory of Biomass and Green Technologies, Gembloux Agro-Bio Tech, University of Liège, 5030 Gembloux, Belgium; 6Department of Public Health and Preventive Medicine, School of Medicine, Jinan University, Guangzhou 510632, China; 7Law School of Artificial Intelligence, Shanghai University of Political Science and Law, Shanghai 201701, China; 8School of Journalism and Communication, Jinan University National Media Experimental Teaching Demonstration Center, Jinan University, Guangzhou 510632, China; 9School of Economic, Guangzhou College of Commerce, Guangzhou 511363, China; 10School of Business, University at Albany, State University of New York, Albany, NY 12222, USA; 11Communication Department, University at Albany, State University of New York, Albany, NY 12222, USA; 12Department of Infectious Diseases and Public Health, Jockey Club College of Veterinary Medicine and Life Science, City University of Hong Kong, Hong Kong SAR, China

**Keywords:** COVID-19, Google trend, health communication, public health measures, panel data analysis

## Abstract

The COVID-19 outbreak at the end of December 2019 spread rapidly all around the world. The objective of this study is to investigate and understand the relationship between public health measures and the development of the pandemic through Google search behaviors in the United States. Our collected data includes Google search queries related to COVID-19 from 1 January to 4 April 2020. After using unit root tests (ADF test and PP test) to examine the stationary and a Hausman test to choose a random effect model, a panel data analysis is conducted to investigate the key query terms with the newly added cases. In addition, a full sample regression and two sub-sample regressions are proposed to explain: (1) The changes in COVID-19 cases number are partly related to search variables related to treatments and medical resources, such as ventilators, hospitals, and masks, which correlate positively with the number of new cases. In contrast, regarding public health measures, social distancing, lockdown, stay-at-home, and self-isolation measures were negatively associated with the number of new cases in the US. (2) In mild states, which ranked one to twenty by the average daily new cases from least to most in 50 states, the query terms about public health measures (quarantine, lockdown, and self-isolation) have a significant negative correlation with the number of new cases. However, only the query terms about lockdown and self-isolation are also negatively associated with the number of new cases in serious states (states ranking 31 to 50). Furthermore, public health measures taken by the government during the COVID-19 outbreak are closely related to the situation of controlling the pandemic.

## 1. Introduction

### 1.1. Public Health Measures in the US

First detected in Wuhan, COVID-19 was highly virulent and spread to Iran, Europe, and finally, North America, contributing to an unprecedented global health crisis [1].

However, among these areas, the coronavirus raging through the US severely affected the country and gradually accounted for the most COVID-19 cases and deaths worldwide [2]. The US is facing a major public health crisis and is trying to address this issue through a series of medical reforms [3].

Since the pandemic, many governments introduced public health measures to control the development of the pandemic and the use of medical substances [4]. Quarantine, social distancing, lockdown, and staying at home were the pivotal role of old-style public health measures in the novel coronavirus (COVID-19) outbreak [5,6]. Quarantine was one of the oldest and most effective tools for controlling communicable disease outbreaks [6]. This public health practice was widely used in fourteenth-century Italy when ships arriving at the port of Venice from plague-infected ports had to anchor and wait for 40 days before disembarking their surviving passengers [7]. Social distancing was designed to reduce interactions between people in a broader community in which individuals may be infectious but have not yet been identified as such and therefore not yet isolated [6]. In South Korea and China social distancing has played an important role in mitigating the spread of COVID-19 [8,9]. In particular, to alert European countries that they should avoid close contact at the individual level and social meetings in each country [10]. The most restrictive nonpharmaceutical interventions (NPIs) for controlling the spread of COVID-19 were mandatory stay-at-home and business closures (“lockdown”) [11]. They were used in some countries to prevent and control the pandemic [12,13,14,15].

### 1.2. Impact of the COVID-19 Pandemic

As a disease caused by a new coronavirus called SARS-CoV-2, COVID-19 affected more than 200 countries and caused respiratory tract infections in humans ranging from mild symptoms to lethal outcomes [16]. The extremely high infection rate and relatively high mortality raised public awareness of the emergence of COVID-19 and hence led to fears, worries, and anxiety among individuals globally. Dealing with COVID-19 had become one of the emergent global health challenges in managing infectious diseases [17,18]. More and more arduous efforts by health experts and authorities were put into mitigating the spread of the virus, and more individuals were beginning to focus on searching for relevant preventive information and taking preventive actions [19].

In less than half a year (in October 2020), 200 countries had reported over 10.3 million confirmed COVID-19 cases worldwide [20], and the cumulative number of cases reported globally was over 323 million; the number of deaths exceeded 5.5 million as of 16 January 2022 [21]. The arrival of the Omicron strain led to the adoption of changes in public safety measures in many areas, but the study of early prevention measures is still relevant. Public safety measures in coping with the coronavirus could significantly ease the spread of the disease and reduce the effects of disease during the pandemic [22,23].

The public searches for medical-related information and safety measures through the internet, where search engines can record the public attention on pandemic-related topics. As one of the largest search engines in the world, Google Trends can reflect public health information-seeking behaviors during the pandemic, especially regarding safety measures in local areas.

Because of the usefulness of public measures for the spread of the disease and the universality of Google search, the objective of in this paper is to investigate and understand the correlation between public health measures and public search behaviors with the development of the disease during the COVID-19 pandemic in the United States.

### 1.3. Aims

The primary aim of this study is to verify whether there is a correlation between Google search query keywords and the development of the COVID-19 pandemic in the United States (research question 1, RQ 1).

The secondary objective is to investigate whether there are significant correlations between the state’s anti-pandemic public health (control) measures and the development of the pandemic (research question 2, RQ 2).

## 2. Materials and Methods 

### 2.1. Keyword Search Tools

Google Trends (https://trends.google.com/trends/ (accessed on 10 April 2020) is a publicly available web-based tool that provides access to a largely unfiltered sample of actual search requests (queries) made to Google. Users could analyze interest in a specific topic from around the globe or drill down to geography at the city level by using its data which are anonymized, categorized, and grouped (support.google.com/trends).

Google search queries had been used to understand health communication during the COVID-19 outbreak [18,24,25,26,27,28]. As the COVID-19 outbreak continued, it was plausible that people might increasingly use internet search engines such as Google search (Google Inc., Mountain View, CA, USA) to obtain information about symptoms, diagnoses, or prevention [29]. Google released these Google COVID-19 search trends as open-access datasets for public health surveillance purposes as well as to accelerate valuable insights into the spread and impact of COVID-19 [30,31,32].

### 2.2. Obtaining and Analysis of the COVID-19 Pandemic Dataset

Google Trend, one of the most popular search engines worldwide, was used in our study to track down the updated internet hit search volumes in different states in the United States. Users of Google Trends could search through the link https://trends.google.com (accessed on 10 April 2020) to obtain the corresponding search volume (SV) of the terms [22] that revealed the general concern of the citizens in a certain area about one topic. We utilized this method in our study for analyzing public concerns related to certain health topics; however, the search volume turned out to be a normalized relative search volume (RSV) instead of an exact search count. The RSV value of relative search volume varies from 0 to 100, where the value of 0 referred to the least popular search term and the value of 100 meant the most popular search term [33,34]. Therefore, we compared and scaled all search information and normalized the data.

According to previous Google search research [35], symptoms, treatments and medical resources, measures, and the virus itself were some of the major concerns reported by online media platforms during the outbreak period of COVID-19. Therefore, in this research, we chose four main topics for search terms: “Diseases,” “Treatments and medical resources,” “Symptoms,” and “Public health measures”. 

### 2.3. Public-Health-Measures-Related Keyword Search

Google search query variables were mainly divided into diseases, symptoms, treatments and medical resources, and public health measures. They were determined according to the pandemic-related search terms mentioned in the relevant literature or the relevant important pandemic research.

It was first called “unknown” virus pneumonia and then it was called “new coronavirus pneumonia”; COVID-19 was introduced by WHO [36], after that, COVID-19 was widely used in the study of this “unknown” virus pneumonia [37,38]. Facing COVID-19, there was presumably no pre-existing immunity in the population against the new coronavirus, and everyone in the population was assumed to be susceptible [37]. Among them, we had chosen pneumonia and COVID-19 to describe the disease. Its symptoms were usually fever, cough, sore throat, breathlessness, fatigue, and malaise among others [38]. Fever was the most critical signal feature of COVID-19 infection and was usually accompanied by fatigue and malaise; the cough usually contained a sore throat, and breathlessness was a serious and obvious symptom [39]. Therefore, we chose fever, cough, and shortness of breath to describe the COVID-19 symptoms.

After COVID-19 had swept the world, the biggest concern was whether the healthcare system could handle a large number of patients. The number of hospitals and doctors was closely related to the control of the pandemic [40]. Ventilators were essential as medical supplies for the treatment of critically ill patients [41]. Masks and vaccines were very effective medical resources for preventing the spread of COVID-19 and had always been widely considered [42,43]. Therefore, we added ventilator, hospital, doctor, vaccine, and mask as the main variables to reflect the treatments. However, many public health measures also played an important role in the control of the spread of COVID-19, such as quarantine, social distancing, lockdown, stay-at-home, etc. [5]. Among them, quarantine and lockdown were general measures implemented by the government, and social distancing and staying at home were voluntary actions of the people [6]. We chose these four search terms to comprehensively discuss public health measures.

### 2.4. Statistical Analysis

According to the above data, we obtained four groups of data related to COVID-19 from google search queries in the United States on COVID-19 from 1 January 2020, to 4 April 2020. We then used random effect analysis to investigate and analyze the relations between the key query terms with the newly added cases.

#### 2.4.1. Panel Unit Root Test

Before the panel dynamic regression model was fitted, the stationary of each time series needed to be tested. If there was a unit root in the autoregressive part of the model, it indicated that this series was not stationary, that is, as time progressed, it did not return to a given value (long-term average), which made the regression a spurious regression. This article employed the Augmented Dickey–Fuller test (ADF test) [44] and the Phillips–Perron test (PP test) [45] to test the stationaries of variables.

For each time series $x_{t}$ (such as positive increase etc.), we structured a p-order autoregressive model AR(p) as follows:$$x_{t}=\phi_{1}x_{t-1}+\cdots \phi_{p}x_{t-p}+\varepsilon_{t},$$
where t=1,⋯,T. Therefore, the ADF test set $\rho =\phi_{1}+\cdots +\phi_{p}-1$ and the null and the alternative hypotheses were as follows:$$\begin{array}{l} H_{0}:\rho =0\;\{x_{t}\mathrm{was}\;a\;\mathrm{non}-\mathrm{stationaryseries}\},\\ H_{1}:\rho <0\;\{x_{t}\mathrm{was}\;a\;\mathrm{stationaryseries}\}. \end{array}$$

The ADF test statistic was:$$\tau =\frac{\hat{\rho }}{S(\hat{\rho })},$$
where $S(\hat{\rho })$ was the sample standard deviation of ρ.

However, the ADF test method had a basic setting in that the noise of the time series had the same variance, so the ADF test method was not effective in the time series with heterogeneity. Phillips and Perron presented the PP test statistics, which can be applied to the stationary test for heterogeneity; it also obeyed the limit distribution of the ADF test statistic. The PP test statistic was:$$Z(\tau )=\tau (\frac{{\overset{⌢}{\sigma }}^{2}}{\sigma_{Sl}^{2}})-\frac{1}{2}(\sigma_{Sl}^{2}-{\overset{⌢}{\sigma }}^{2})T\sqrt{\sigma_{Sl}^{2}\sum_{t=2}^{T}{\left(x_{t-1}-{\bar{x}}_{T-1}\right)}^{2}},$$
where ${\overset{⌢}{\sigma }}^{2}=T^{-1}\sum_{t=1}^{T}{\overset{⌢}{\varepsilon }}_{t}^{2}$,${\overset{⌢}{\sigma }}_{Sl}^{2}=T^{-1}\sum_{t=1}^{l}{\overset{⌢}{\varepsilon }}_{t}^{2}+2T^{-1}\sum_{j=1}^{t}w_{j}(l)\sum_{t=j+1}^{T}{\overset{⌢}{\varepsilon }}_{t}{\overset{⌢}{\varepsilon }}_{t-j}$,${\bar{x}}_{T+1}={\left(T-1\right)}^{-1}$$\sum_{t=1}^{T-1}x_{t}$. We performed a panel unit root test at a given confidence level; the values 0.001, 0.01, and 0.05 for the confidence level were the most frequently used [44,45].

#### 2.4.2. Panel Dynamic Regression Model

In order to estimate a positive increase by the panel data analysis, it was common to explain the non-observed heterogeneity by using a fixed or random effect. The general form of the panel dynamic regression model was as follows [46]:$$y_{it}=\alpha +\sum_{g=1}^{G}\beta_{g}x_{git}+u_{it}+\varepsilon_{it},\;(1)$$
where $y_{it}$ was the response (that was a positive increase); α was the intercept term (a constant); $\beta =\left(\beta_{1},\ldots ,\beta_{G}\right)$ were the parameters to be estimated; $x_{it}=\left(x_{1it,\ldots ,}x_{git}\right)$ were the predictors; $x_{git}$ was the g-th predictor for the individual i into the observation value at period t; and $u_{it}$ was an individual heterogeneity, which was non-observed. i=1,⋯,I, t=1,⋯,T.

In the fixed effect model, the individual heterogeneity $u_{it}$ was as a constant; so we can rewrite the Model (1) as:$$y_{it}=\theta +\sum_{g=1}^{G}\beta_{g}x_{git}+\varepsilon_{it},\;(2)$$
where $\theta =\alpha +u_{it}$.

On the contrary, the random effect model assumed $u_{it}$ as a random variable and rewrote Model (1) as:$$y_{it}=\alpha +\sum_{g=1}^{G}\beta_{g}x_{git}+\delta_{it},\;(3)$$
where $\delta_{it}=u_{it}+\varepsilon_{it}$.

Before performing the panel data regression analysis, it was necessary to choose a fixed effect model (2) or a random effect model (3) to fit the model according to the specific actual data. If the model selection was wrong, it would cause a huge estimation error and fail to analyse [47].

#### 2.4.3. Hausman Test

In order to choose a fixed effect model (2) or a random effect model (3), we conducted a Hausman test [46]. This is because the Hausman test would lose more degrees of freedom when estimating a fixed effect model, and the Hausman test tests whether a random effect model would be appropriate. The Hausman test set the null and the alternative hypotheses as follows:$$\begin{array}{l} H_{0}:\mathrm{cov}(x_{it},u_{it})=0\;\{u_{it}\;\mathrm{was}\;a\;\mathrm{random}\;\mathrm{variable}\},\\ H_{1}:\mathrm{cov}(x_{it},u_{it})\neq 0\;\{u_{it}\;\mathrm{was}\;a\;\mathrm{constant}\}. \end{array}$$

The Hausman test statistics were:$$W={\left[{\overset{⌢}{\beta }}_{fe}-{\overset{⌢}{\beta }}_{re}\right]}^{Τ}{\overset{⌢}{\Omega }}^{-1}[{\overset{⌢}{\beta }}_{fe}-{\overset{⌢}{\beta }}_{re}],$$
where ${\overset{⌢}{\beta }}_{fe}$ and ${\overset{⌢}{\beta }}_{re}$ were the estimated results of the regression parameters β in the fixed effect model (2) and random effect model (3), respectively, and $\overset{⌢}{\Omega }=Var({\overset{⌢}{\beta }}_{fe}-{\overset{⌢}{\beta }}_{re})$.

We performed a Hausman test at a given confidence level, which commonly used 0.001, 0.01, and 0.05 [46].

## 3. Result

In total, we collected and normalized all states’ Google search query data in the United States from 1 January 2020 to 4 April 2020 and saved it into CSV files. We also added CDC newly added cases data for coronavirus from CDC into the dataset [48]. Variables and sources and the basic descriptive statistics are listed in Table 1 below.

### 3.1. Full Sample Empirical Regression

In Model (1), I was the number of states in the United States, I=50; T was the number of days from 1 January 2020 to 4 April 2020, T=95; G was the number of predictors, G=15; the unit root test results for each variable are shown in Table 2; and the Hausman test result of the full sample is shown in Table 3.

Table 2 presents the results of the two-unit root tests for each variable. If the sample sequence were non-stationary, it would need to be processed by a difference or a lag operator to make it a stationary time series. However, both the ADF test and PP test results for 15 variables rejected the null hypothesis at the 0.1% level; all 15 variables were stationary.

After the stationary test, the second step was to apply a Hausman test to assess whether the panel data Model (1) was a fixed or a random effect. The Hausman test result was shown in Table 3. The result showed that the *p*-value is 0.994, which meant the null hypothesis of the Hausman test was not rejected at the 0.1%, 1%, and 5% levels; we chose a random effect model (e.g., Model (3)) to analyze the full sample.

Based on the result of the Hausman test, the random effect model was selected for the full sample estimation in this study. The estimation results are shown in Table 4.

To answer RQ1, we listed different Google search query keywords with their random effect results of the full sample of Model (3) in Table 4.

With concern about changing the name from coronavirus to COVID-19, only pneumonia (2.11) was found to have a positive coefficient with newly added cases for disease name-related search queries.

In terms of symptom-related queries, fever was found to have a significant positive coefficient (1.972), which indicated that the most significant symptom for predicting infection was high temperature. Others were not significant, such as shortness of breath (1.576).

For searches related to treatments and medical resources, positive correlations were found between infected cases and searches for a ventilator (8.424), hospital (0.77), and mask (0.574). This meant that more people searching for medical resources of these kinds could predict the increase in the number of positive cases. This was a warning sign to authorities that medical resources such as masks, hospitals, and especially ventilators could be generally insufficient.

With public-health-measures-related search terms, quarantine had a significant positive coefficient (0.587), while lockdown (−2.172) and self-isolation (−6.25) had significant negative coefficients. Public health measures towards passive individual control such as quarantine could positively predict newly added cases, which meant that when citizens searched for public controls at the passive individual level, infections increased rapidly. Compared to collective-level search terms, such as lockdown, the coefficient was negative, which meant that when people noticed a lockdown in an area, it was associated with a decrease in new cases. Search terms on social distancing and self-isolation were significantly associated with a drop in newly added COVID-19 cases. Relevant medical management departments needed to increase public concern about preventive measures at the individual active/initiative level and further carried out collective level control measures, while the effect of passive individual control measures may not be significant.

The results showed that to answer RQ1, the increase or decrease in COVID-19 cases was partly related to these search variables in the United States on the Google platform. The query terms related to treatments and medical resources, ventilator, hospital, and mask were positive predictors of COVID-19 cases. This was probably because COVID-19 was the most harmful to the respiratory system and ventilators were very expensive and scarce. In terms of symptoms, people could find and compare symptoms online to see whether they have COVID-19. This allowed potential patients to be detected and confirmed as early as possible. They could thereby reduce travel, ensure safe social distancing and self-isolate, and seek medical treatment at hospitals. Public safety measures such as lockdown and self-isolation were negative predictors of the number of new cases, reflecting that these measures could have reduced the number of new cases.

### 3.2. Sub-Sample Empirical Regression

To answer RQ2, we discussed the impact of public health measures between states, this study collected data from 50 states in the United States, focusing on the effect of the significant variables (quarantine, lockdown, and self-isolation) of public health measures in Table 4 on the increase in COVID-19 cases.

This article defined the first day with a non-zero increase in new cases as the day when the outbreak period began and set the number of days in the outbreak period in each state as $H_{i}$; the sum of daily positive increase of each state was $L_{i}$ and the average daily positive increase during the outbreak period was $K_{i}$, where: $K_{i}=L_{i}/H_{i}$, i=1,…,50. Additionally, this study used the average daily positive increase $K_{i}$ to sort 50 states from least to most, as shown in Table 5.

As shown in Table 5, we found that the average daily positive increase $K_{i}$ of the state of Alaska was the lowest, while the average daily positive increase $K_{i}$ of New York was the highest in the 50 states of the United States. To better discuss the 50 states of the United States, we divided them into five categories according to the state rankings and calculated the mean values of $H_{i}$ (i.e., the number of days in the outbreak period) and $K_{i}$ (i.e., the average daily positive increase during the outbreak period) in each category, as shown in Table 6.

According to Table 6, both the mean values of $H_{i}$ (i.e., the number of days in the outbreak period) and $K_{i}$ (i.e., the average daily positive increase during the outbreak period) presented a gradual increase in the five categories. Additionally, the mean values of $H_{i}$ were between 24–30 days in each category but the mean values of $K_{i}$ were quite different, with the lowest being 11.648 and the highest being 793.319. Then, we used Model (1) to perform a panel data regression on the top 20 (Alaska to Oregon) and the last 20 cities (Missouri to New York); the Hausman test results are shown in Table 7.

Table 7 presents the Hausman tests for assessing whether the two sub-sample panel data models were fixed effect models or random effect models. The results showed that the *p*-value of states ranking 1 to 20 was 1.000, and the *p*-value of states ranking 31 to 50 was 0.846, which meant the null hypotheses of the two Hausman tests were not rejected at the 0.1%,1%, and 5% levels. Based on the results of the Hausman tests, the random effect model was selected for both sub-sample estimations in this study. The partial estimation results of the significant variables of public health measures in full sample regression were shown in Table 8 and Table 9, respectively.

Table 8 shows that in the states ranking 1 to 20, with public-health-measures-related search terms, quarantine (−0.024), lockdown (−0.035), and self-isolation (−0.185) have negative coefficients, and the significant variables were the same as the full sample. It was particularly important to note that search queries such as quarantine and self-isolation were individual passive and active behaviors in public health measures, while lockdown was an overall government-imposed or suggested public health measure, which meant that in mild states (states ranking 1 to 20), such as Alaska, North Dakota, Wyoming, South Dakota, Montana and so on, all public health measures search queries (i.e., quarantine, lockdown, and self-isolation) were significantly associated with the reduction of COVID-19 situations.

However, as Table 9 shows, in the states ranking 31 to 50, the variables lockdown (−3.208) and self-isolation (−20.735) had significant negative coefficients but the quarantine was not significant at 0.1%, 1%, and 5% levels, which meant in serious states (states ranking 31 to 50), such as Washington, Illinois, Pennsylvania, Florida, Massachusetts and so on, only government-imposed or suggested quarantine and self-isolation search queries were significantly associated with the reduction of COVID-19 situations; especially the self-isolation search query had a large negative correlation to COVID-19 cases.

To answer RQ2, the searching behaviors suggested that the public health measures taken by the government in response to COVID-19 were closely related to the situation of the pandemic. In mild states (states ranking 1 to 20), all public health measures search queries were significantly associated with the reduction of COVID-19 situations. In serious states (states ranking 31 to 50), only quarantine and self-isolation search queries were significantly associated with the reduction of COVID-19 situations; especially the self-isolation search query had a large negative correlation to COVID-19 cases.

## 4. Discussion

### 4.1. Interpretation of the Results

Given the continuing pandemic, Google search keywords could reflect the trend of health information that the public seeks. Thus, we discussed the correlation between public health measures and public search behaviors with the development of the disease during the COVID-19 pandemic in the United States using panel data analysis. The four main query topics we chose—“Diseases,” “Treatments and medical resources,” “Symptoms,” and “Public health measures”—represented people’s major concern for the outbreak during search activities with 3–5 query variables each. We tested for the stationarity of data and found that all variables were stationary (tested by ADP test and PP test). Applying the Hausman test we also found that the random effect model is appropriate for our data.

The result showed that the search queries were related to the development of the disease. As we adopted all samples, the search query terms had an increase or a decrease in COVID-19 cases at different levels. Search query terms that had a positive correlation with the COVID-19 pandemic included ventilator, hospital, and mask, and pneumonia, fever, quarantine, lockdown, and self-isolation had a negative correlation with the COVID-19 pandemic. For further study on the impact of the state’s anti-pandemic public health measures on the COVID-19 pandemic, we ranked the 50 states by the average daily positive increase from lowest to highest. The panel data analysis of the mild states (states ranking 1 to 20) showed that the significant variables were the same with the full sample (i.e., quarantine, lockdown, and self-isolation), but quarantine search query terms had a negative relationship with COVID-19 pandemic. However, quarantine search query terms had no relationship with the COVID-19 pandemic in the serious states (states ranking 31 to 50), the analysis of lockdown and self-isolation still had a negative correlation with the COVID-19 pandemic. Comparing the results of the full sample and two sub-samples, we found that lockdowns and self-isolation related to COVID-19 cases negatively and significantly. Quarantine had a positive coefficient in full sample regression but had a negative coefficient in mild states regression and was not significant in serious states regression. This meant that policies might increase health communication and discussions on the measures such as quarantine, lockdown, and self-isolation to confront COVID-19.

### 4.2. Future Perspectives

Appropriate public health measures can limit the spread of COVID-19. The symptoms of COVID-19 are similar to the flu (such as fever, cough, or sore throat). COVID-19 occurs during respiratory disease season in America and its symptoms such as fever or feeling feverish/chills, cough, and shortness of breath are similar to influenza. In addition, the incidence of the disease in each state is mainly related to the sequence of the disease. States with a later onset of the disease can learn from the experience of states with an earlier onset of the disease and adopt better isolation and lockdown measures. Therefore, the government could take more and better measures to control the spread of the outbreak (e.g., all people aged 6 months or older should be vaccinated annually) [49,50,51]. Additionally, in a study based on the data of 1700 locations deployed during the pandemic across China, Korea, Italy, Iran France, and the United States, as well as national non-drug interventions, researchers show that the measures have prevented or delayed the 6.1 million confirmed cases and avoided about 495 million infectious cases [52]. Proper compulsory public health measures can avoid infection to a certain degree [53,54]. The government should suggest effective public measures according to its national conditions to cope with coronavirus as a disaster that challenges mankind.

### 4.3. Limitations

This study has certain limitations. First, we only used the Google search engine, and the scope of the query, therefore, is limited to the United States and English language users; therefore, can’t fully reflect people’s choices in other countries or regions of the world. In addition, the Google search users’ demographics in our study such as age and gender are not available, and hence we cannot do further analysis. Moreover, the effect of these prevention and control measures is related to the original strain and Delta strain in the earlier period. If it is an Omicron strain, the effect of the prevention and control measures is almost ineffective. Future studies could be improved in these aspects.

## 5. Conclusions

In this study of the US Google search query sample, public search behaviors and the development of the COVID-19 pandemic are analyzed and found to be related. Public-health-measure searches, including quarantines, lockdowns, and self-isolation, are negatively correlated to outbreaks with varying degrees across states. These results underscore the need for governments to take more and better health measures to mitigate the development of COVID-19.

## Figures and Tables

**Table 1 ijerph-20-03007-t001:** Descriptive statistics of the main variables.

Source	Search Query Topics	Variable	Mean	Std. Dev.	Min	Max
CDC		Positive increase	64.280	464.245	0	10,841
Search query	Diseases	Pneumonia	19.200	11.043	0.1	124
		COVID-19	114.700	212.172	0.1	1400
	Symptoms	Shortness of breath	4.382	5.517	0.1	74
		Fever	43.790	20.768	0.1	200
		Cough	33.340	15.057	0.1	200
	Treatments and medical resources	Ventilator	7.628	13.781	0.1	180
		Hospital	194.800	65.158	0.1	500
		Vaccine	31.210	22.777	0.1	216
		Doctor	86.020	27.849	0.1	230
		Mask	132.800	164.294	0.1	2500
	Public health measures	Quarantine	56.180	70.950	0.1	700
		Social distance	2.143	4.786	0.1	48
		Lockdown	23.260	38.305	0.1	400
		Stay at home	24.660	88.260	0.1	1600

**Table 2 ijerph-20-03007-t002:** Unit root test results for 15 variables.

	Variable	ADF Test	PP Test
	Positive increase	−13.080 ***	−517.240 ***
Diseases	Pneumonia	−12.160 ***	−5595.000 ***
	COVID-19	−14.940 ***	−351.900 ***
Symptoms	Shortness of breath	−13.283 ***	−5307.900 ***
	Fever	−14.181 ***	−3010.200 ***
	Cough	−12.253 ***	−5861.100 ***
Treatments and medical resources	Ventilator	−15.715 ***	−1598.800 ***
	Hospital	−6.4026 ***	−2950.300 ***
	Vaccine	−10.325 ***	−2662.900 ***
	Doctor	−7.7911 ***	−4502.700 ***
	Mask	−13.789 ***	−1667.200 ***
Public health measures	Quarantine	−15.639 ***	−917.420 ***
	Social distance	−13.856 ***	−3297.300 ***
	Lockdown	−14.085 ***	−1235.600 ***
	Stay at home	−15.909 ***	−2103.500 ***
	Self-isolation	−12.574 ***	−4659.600 ***

***: p<0.001.

**Table 3 ijerph-20-03007-t003:** Hausman test result of the full sample of Model (1).

Sample Type	Hausman Statistic	*p*-Value
Full sample	4.799	0.994

**Table 4 ijerph-20-03007-t004:** Random effect results of the full sample of Model (3).

	Variable	Coef.	z-Value	Pr (>|z|)
	Constant	−315.677	−8.355	0.000 ***
Diseases	Pneumonia	2.110	3.670	0.000 ***
	COVID-19	−0.097	−1.522	0.128
Symptoms	Shortness of breath	1.576	1.360	0.174
	Fever	1.972	5.029	0.000 ***
	Cough	0.543	1.197	0.231
Treatments and medical resources	Ventilator	8.424	12.750	0.000 ***
	Hospital	0.770	6.046	0.000 ***
	Vaccine	−0.433	−1.283	0.199
	Doctor	−0.080	−0.330	0.742
	Mask	0.574	11.474	0.000 ***
Public health measures	Quarantine	0.587	3.369	0.000 ***
	Social distance	−2.697	−1.655	0.098
	Lockdown	−2.172	−8.415	0.000 ***
	Stay at home	−0.067	−0.846	0.398
	Self-isolation	−6.250	−2.507	0.012 *

***: p<0.001; *: p<0.05.

**Table 5 ijerph-20-03007-t005:** State rankings of the average daily positive increase ($K_{i}$).

Ranking	State	Ranking	State	Ranking	State
1	Alaska	18	Minnesota	35	Ohio
2	North Dakota	19	Arkansas	36	Colorado
3	Wyoming	20	Oregon	37	Indiana
4	South Dakota	21	Oklahoma	38	Connecticut
5	Montana	22	Idaho	39	Texas
6	Nebraska	23	Utah	40	Georgia
7	Hawaii	24	Mississippi	41	Washington
8	West Virginia	25	South Carolina	42	Illinois
9	Vermont	26	Arizona	43	Pennsylvania
10	Maine	27	Alabama	44	Florida
11	New Hampshire	28	Nevada	45	Massachusetts
12	New Mexico	29	North Carolina	46	California
13	Delaware	30	Wisconsin	47	Louisiana
14	Rhode Island	31	Missouri	48	Michigan
15	Kansas	32	Virginia	49	New Jersey
16	Kentucky	33	Maryland	50	New York
17	Iowa	34	Tennessee		

**Table 6 ijerph-20-03007-t006:** Mean of the number of days in the outbreak period ($H_{i}$) and the average daily positive increase ($K_{i}$).

	$\mathrm{Mean}\;\mathrm{of}\;H_{i}$	$\mathrm{Mean}\;\mathrm{of}\;K_{i}$
States ranking 1 to 10	24.6	11.648
States ranking 11 to 20	27.7	26.498
States ranking 21 to 30	26.1	63.907
States ranking 31 to 40	28.6	141.366
States ranking 41 to 50	29.8	793.319

**Table 7 ijerph-20-03007-t007:** Hausman test results of two sub-samples of Model (1).

Sample Type	Hausman Statistic	*p*-Value
States ranking 1 to 20	1.008	1.000
States ranking 31 to 50	9.569	0.846

**Table 8 ijerph-20-03007-t008:** Random effect partial results of states ranking 1 to 20.

	Variable	Coef.	z-Value	Pr (>|z|)
Public health measures	Quarantine	−0.024	−3.514	0.000 ***
	Lockdown	−0.035	−2.918	0.004 **
	Self-isolation	−0.185	−2.272	0.023 *

***: p<0.001; **: p<0.01; *: p<0.05.

**Table 9 ijerph-20-03007-t009:** Random effect partial results of states ranking 31 to 50.

	Variable	Coef.	z-Value	Pr (>|z|)
Public health measures	Quarantine	0.244	0.577	0.564
	Lockdown	−3.208	−5.644	0.000 ***
	Self-isolation	−20.735	−2.383	0.017 *

***: p<0.001; *: p<0.05.

## Data Availability

The data used in this study is available upon request from the corresponding author.

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
