# Peer review of "COVID-19 Related Early Google Search Behavior and Health Communication in the United States: Panel Data Analysis on Health Measures"

_ijerph, 2023, doi:10.3390/ijerph20043007_

Round 1
Reviewer 1 Report (Previous Reviewer 2)
In the opinion of the reviewer, the article after the corrections is suitable for publication.
Author Response
Re: COVID-19 related early google search behavior and health communication in the United States: Panel Data Analysis on health measures
(Ref. ID ijerph-2153977)
Dear Editors and reviewers,
We are deeply grateful to the Editor, the Associate Editor and the reviewers for their constructive comments and suggestions, which have helped significantly in improving the revised manuscript COVID-19 related early google search behavior and health communication in the United States: Panel Data Analysis on health measures (Ref. ID ijerph-2153977).
We have revised the paper and would like to re-submit it for your consideration. We have carefully read all the comments and addressed all issues raised by the reviewer as follows. All the responses are in red color and the changes made within the manuscript are in blue color. I look forward to hearing from you soon.
Best regards,
Qian Liu
Associate Professor, Jinan University
Response to Reviewer 1 Comments
Point 1: In the opinion of the reviewer, the article after the corrections is suitable for publication.
Response 1: Thank you very much for your positive comment!

Reviewer 2 Report (New Reviewer)
Dear authors,
Thank you for the opportunity to review this interesting study of the relationship between Google search terms and the development of the Covid-19 pandemic, and related government measures in the USA. This is overall a clearly structured paper that presents interesting analyses on a relevant topic.
While most of the methodological choices are clearly presented and argumented for, it is less clear how the chosen search terms - which are the focus of the study - were selected. Reference is made to terms arising from previous research, but this research is not clearly referenced. Furthermore, it is unclear whether certain terms were excluded or considered less relevant, and if so why. Ultimately, it should be clearer why the search terms selected were the most appropriate and valid for this particular study.
The authors are inconsistent in their use of the terms epidemic and pandemic. The correct term is most likely pandemic but this should be checked and corrected throughout.
Stylistically, at some points throughout the paper the authors seem to be writing at an earlier phase of the pandemic. Perhaps this could be addressed by placing the timeframe of analysis and reference to that clearly in the past tense?
Some of the recommendations made, for example the last sentence in the abstract, are not clearly connected to the analysis and discussion presented in the paper. This could be improved.
Author Response
Re: COVID-19 related early google search behavior and health communication in the United States: Panel Data Analysis on health measures
(Ref. ID ijerph-2153977)
Dear Editors and reviewers,
We are deeply grateful to the Editor, the Associate Editor and the reviewers for their constructive comments and suggestions, which have helped significantly in improving the revised manuscript COVID-19 related early google search behavior and health communication in the United States: Panel Data Analysis on health measures (Ref. ID ijerph-2153977).
We have revised the paper and would like to re-submit it for your consideration. We have carefully read all the comments and addressed all issues raised by the reviewer as follows. All the responses are in red color and the changes made within the manuscript are in blue color. I look forward to hearing from you soon.
Best regards,
Qian Liu
Associate Professor Jinan University
Response to Reviewer 2 Comments
Point 1: Thank you for the opportunity to review this interesting study of the relationship between Google search terms and the development of the Covid-19 pandemic, and related government measures in the USA. This is overall a clearly structured paper that presents interesting analyses on a relevant topic.
Response 1: Thank you very much for your positive comment!
Point 2: While most of the methodological choices are clearly presented and argumented for, it is less clear how the chosen search terms - which are the focus of the study - were selected. Reference is made to terms arising from previous research, but this research is not clearly referenced. Furthermore, it is unclear whether certain terms were excluded or considered less relevant, and if so why. Ultimately, it should be clearer why the search terms selected were the most appropriate and valid for this particular study.
Response 2: Thank you very much for your comment!
- In order to explain how the search terms are chosen during the time “from January 1 to April 4, 2020”, we illustrate the importance of COVID-19 in subsection 2.3 as
“First called “unknown” virus pneumonia and then it was called “New coronavirus pneumonia” and COVID-19 was introduced by WHO[36], after that, COVID-19 was widely used in the study of this “unknown” virus pneumonia[37.38]. ”
Please see lines 147-149 of page 3.
- According to your suggestion, we supplement the reasons for choosing “fever, cough, and shortness of breath” to describe the COVID-19 symptoms and ignoring other search words in subsection 2.3 as:
”Its symptoms were usually fever, cough, sore throat, breathlessness, fatigue, and malaise among others[38]. Because fever was the most critical signal feature of COVID-19 infection and was usually accompanied by fatigue and malaise, the cough usually contained a sore throat, and breathlessness was a serious and obvious symptom[39]. So, we chose fever, cough, and shortness of breath to describe the COVID-19 symptoms.”
Please see lines 152-157 of page 4.
- We add the explanation for selecting “quarantine, social distancing, lockdown and stay at home” to represent the public health measures in subsection 2.3 as:
“So, we added ventilator, hospital, doctor, vaccine, and mask as the main variables to reflect the treatments. However, many public health measures also played an important role in the control of the spread of COVID-19, such as quarantine, social distancing, lockdown, stay at homeetc[5]. Among them, quarantine and lockdown were general measures implemented by the government, and social distancing and staying at home were voluntary actions of the people[6]. We chose these four search terms to comprehensively discuss public health measures. ”
Please see lines 163-169 of page 4.
Point 3: The authors are inconsistent in their use of the terms epidemic and pandemic. The correct term is most likely pandemic but this should be checked and corrected throughout.
Response 3: Thank you very much for your comment!
According to your suggestion, we changed “epidemic” to “pandemic” in the manuscript, 26 places in total.
Point 4: Stylistically, at some points throughout the paper the authors seem to be writing at an earlier phase of the pandemic. Perhaps this could be addressed by placing the timeframe of analysis and reference to that clearly in the past tense?
Response 4: Thank you very much for your comment!
According to your suggestion, we revise the references in the past tense in subsections 1.1, 1.2, 2.1, 2.2, and 2.3. In addition, we rewrite the analysis in the past tense in subsections 2.4, 4.1, and section 3.
Point 5: Some of the recommendations made, for example, the last sentence in the abstract, are not clearly connected to the analysis and discussion presented in the paper. This could be improved.
Response 5: Thank you very much for your comment!
According to your suggestion, we deleted the last sentence which is not clearly connected to the analysis in the abstract.

Reviewer 3 Report (New Reviewer)
1. The incidence of the disease in each state is mainly related to the sequence of the disease. States with the onset of the disease later can learn from the experience of states with the onset of the disease earlier and adopt better isolation and lockdown measures.
2. The effect of these prevention and control measures is related to the original strain and Delta strain in the earlier period. If it is Omicron strain, the effect of the prevention and control measures is almost ineffective.
3. The prevalence of COVID-19 is known by the volume of search queries, which has little to do with the prevention and control of the disease, and the prevalence of COVID-19 can be more accurately obtained through other means.
4. Lines 238-282 are missing.
Author Response
Re: COVID-19 related early google search behavior and health communication in the United States: Panel Data Analysis on health measures
(Ref. ID ijerph-2153977)
Dear Editors and reviewers,
We are deeply grateful to the Editor, the Associate Editor and the reviewers for their constructive comments and suggestions, which have helped significantly in improving the revised manuscript COVID-19 related early google search behavior and health communication in the United States: Panel Data Analysis on health measures (Ref. ID ijerph-2153977).
We have revised the paper and would like to re-submit it for your consideration. We have carefully read all the comments and addressed all issues raised by the reviewer as follows. All the responses are in red color and the changes made within the manuscript are in blue color. I look forward to hearing from you soon.
Best regards,
Qian Liu
Associate Professor Jinan University
Response to Reviewer 3 Comments
Point 1: The incidence of the disease in each state is mainly related to the sequence of the disease. States with the onset of the disease later can learn from the experience of states with the onset of the disease earlier and adopt better isolation and lockdown measures.
Response 1: Thank you very much for the reviewer’s comment!
We add this discovery in subsection 4.2 Future Perspectives as
“In addition, the incidence of the disease in each state is mainly related to the sequence of the disease. States with the onset of the disease later can learn from the experience of states with the onset of the disease earlier and adopt better isolation and lockdown measures. ”
Please see lines 430-433 of page 11.
Point 2: The effect of these prevention and control measures is related to the original strain and Delta strain in the earlier period. If it is Omicron strain, the effect of the prevention and control measures is almost ineffective.
Response 2: Thank you very much for the reviewer’s comment!
We add this limitation in subsection 4.3 Limitations as
“Moreover, the effect of these prevention and control measures is related to the original strain and Delta strain in the earlier period. If it is an Omicron strain, the effect of the prevention and control measures is almost ineffective.”
Please see lines 448-451 of page 11.
Point 3: The prevalence of COVID-19 is known by the volume of search queries, which has little to do with the prevention and control of the disease, and the prevalence of COVID-19 can be more accurately obtained through other means.
Response 3: Thank you very much for the reviewer’s comment!
Since this manuscript is to study the relevance of Google search behavior and health communication under COVID-19, we hope to highlight that COVID-19 is the disease scope of this manuscript. And after COVID-19 was introduced by WHO, COVID-19 was widely used in the study of new pneumonia. Also, we try to compare different independent variables which could be made by the comparison of coefficients. So, we choose COVID-19 as one of the representative search terms to describe the disease and add the reason in subsection 2.3 as:
“First called “unknown” virus pneumonia and then it was called “New coronavirus pneumonia” and COVID-19 was introduced by WHO[36], after that, COVID-19 was widely used in the study of this “unknown” virus pneumonia[37.38].”.
Please see lines 147-149 of page 3.
Point 4: Lines 238-282 are missing.
Response 4: Thank you very much for the reviewer’s comment!
Sorry for this mistake, we complement lines 238-282 as
“3. Result
In total, we collected and normalized all states’ google search query data in United State from January 1, 2020, to April 4, 2020, and saved it into CSV files. We also added CDC newly added cases data for coronavirus from CDC into the dataset[48]. Variables and sources and the basic descriptive statistic are listed in Table 1 below.
Table 1. Descriptive statistics of the main variables.
Source |
Search query topics |
Variable |
Mean |
Std.Dev. |
Min |
Max |
CDC |
|
Positive increase |
64.280 |
464.245 |
0 |
10841 |
Search query |
Diseases |
Pneumonia |
19.200 |
11.043 |
0.1 |
124 |
COVID-19 |
114.700 |
212.172 |
0.1 |
1400 |
||
Symptoms |
Shortness of breath |
4.382 |
5.517 |
0.1 |
74 |
|
Fever |
43.790 |
20.768 |
0.1 |
200 |
||
Cough |
33.340 |
15.057 |
0.1 |
200 |
||
Treatments and medical resources |
Ventilator |
7.628 |
13.781 |
0.1 |
180 |
|
Hospital |
194.800 |
65.158 |
0.1 |
500 |
||
Vaccine |
31.210 |
22.777 |
0.1 |
216 |
||
Doctor |
86.020 |
27.849 |
0.1 |
230 |
||
Mask |
132.800 |
164.294 |
0.1 |
2500 |
||
Public health measures |
Quarantine |
56.180 |
70.950 |
0.1 |
700 |
|
Social distance |
2.143 |
4.786 |
0.1 |
48 |
||
Lockdown |
23.260 |
38.305 |
0.1 |
400 |
||
Stay at home |
24.660 |
88.260 |
0.1 |
1600 |
3.1. Full Sample Empirical Regression
In Model (1), was the number of states in the United, ; was the number of days from January 1, 2020, to April 4, 2020, ; was the number of predictors, , and the unit root test results for each variable were shown in Table 2, the Hausman test result of full sample was shown in Table 3.
Table 2. Unit root test results for 15 variables.
|
Variable |
ADF test |
PP test |
|
Positive increase |
-13.080*** |
-517.240*** |
Diseases |
Pneumonia |
-12.160*** |
-5595.000*** |
COVID-19 |
-14.940*** |
-351.900*** |
|
Symptoms |
Shortness of breath |
-13.283*** |
-5307.900*** |
Fever |
-14.181*** |
-3010.200*** |
|
Cough |
-12.253*** |
-5861.100*** |
|
Treatments and medical resources |
Ventilator |
-15.715*** |
-1598.800*** |
Hospital |
-6.4026*** |
-2950.300*** |
|
Vaccine |
-10.325*** |
-2662.900*** |
|
Doctor |
-7.7911*** |
-4502.700*** |
|
Mask |
-13.789*** |
-1667.200*** |
|
Public health measures |
Quarantine |
-15.639*** |
-917.420*** |
Social distance |
-13.856*** |
-3297.300*** |
|
Lockdown |
-14.085*** |
-1235.600*** |
|
Stay at home |
-15.909*** |
-2103.500*** |
|
Self-isolation |
-12.574*** |
-4659.600*** |
***: .
Table 2 presented the results of two unit root tests for each variable. If the sample sequence was non-stationary, it needed to be processed by a difference or a lag operator to make it a stationary time series. However, both the ADF test and PP test results for 15 variables rejected the null hypothesis at the 0.1% level, all 15 variables were stationary.
Table 3 Hausman test result of the full sample of Model (1).
Sample type |
Hausman statistic |
p-value |
Full sample |
4.799 |
0.994 |
After the stationary test, the second step was to apply a Hausman test to assess whether the panel data Model (1) was a fixed or a random effect. Hausman test result was shown in Table 3. The result showed that the p-value is 0.994, which meant the null hypothesis of the Hausman test was not rejected at the 0.1%, 1%, 5% levels, and we chose a random effect model (e.g. Model (3)) to analyze the full sample.
Based on the result of Hausman test, the random effect model was selected for the full sample estimation in this article. The estimation results were shown in Table 4.
Table 4 Random effect results of the full sample of Model (3).
|
Variable |
Coef. |
z-value |
Pr(>|z|) |
|
Constant |
-315.677 |
-8.355 |
0.000*** |
Diseases |
Pneumonia |
2.110 |
3.670 |
0.000*** |
COVID-19 |
-0.097 |
-1.522 |
0.128 |
|
Symptoms |
Shortness of breath |
1.576 |
1.360 |
0.174 |
Fever |
1.972 |
5.029 |
0.000*** |
|
Cough |
0.543 |
1.197 |
0.231 |
|
Treatments and medical resources |
Ventilator |
8.424 |
12.750 |
0.000*** |
Hospital |
0.770 |
6.046 |
0.000*** |
|
Vaccine |
-0.433 |
-1.283 |
0.199 |
|
Doctor |
-0.080 |
-0.330 |
0.742 |
|
Mask |
0.574 |
11.474 |
0.000*** |
|
Public health measures |
Quarantine |
0.587 |
3.369 |
0.000*** |
Social distance |
-2.697 |
-1.655 |
0.098 |
|
Lockdown |
-2.172 |
-8.415 |
0.000*** |
|
Stay at home |
-0.067 |
-0.846 |
0.398 |
|
Self-isolation |
-6.250 |
-2.507 |
0.012* |
***: ; * : .
To answer RQ1, we listed different Google search query keywords with their random effect results of the full sample of Model (3) in Table 4.
With concern about changing the name from coronavirus to COVID-19, only pneumonia (2.11) was found with a positive coefficient with newly added cases for disease name-related search queries.
In terms of symptom-related queries, fever was found with a significant positive coefficient (1.972), which indicated that the most significant symptom for predicting infection was high temperature. And others were not significant, like shortness of breath (1.576). ”
Please see lines 238-282 of pages 5-7.

This manuscript is a resubmission of an earlier submission. The following is a list of the peer review reports and author responses from that submission.
Round 1
Reviewer 1 Report
Dear authors, I read again and again the paper. every time I get consufeswed and cannot figure how you assess "the correlation between public health measures and public search behaviors with 60 the development of the disease during COVID-19 epidemic in United States." no data collected to support that, no quantitive analysis, etc.
I got the feeling, that many people work separatly on the paper and insert secrtions which are not tied to each other.
My humble recommendation is to:
1. be clear on what you want to report
2. clear objective definition
3. provide hypotheses and verify them
4. Use appropriate data analysis methods (anova, aconva, etc.)
5. Discuss your results
6. Conclude
Your introduction is empty and says nothing.
Sorry for my comments.
Reviewer 2 Report
The amended article is structured in its content and related to the adopted topic.
In the opinion of the reviewer, references (asterisks) to the data in tables 2-4 are incorrectly used, and thus the legend to the data included in the tables is incorrectly prepared in the form of three levels of significance. The authors mainly use only one level (0.001). References to data should be made only in the case of a different assessment than that adopted in the article (0.001), for example in Table 4 in relation to the value of 0.012 at the level of 0.01. These notes also apply to the legends of other tables.